# Water-Related Variables for Predicting Yield of Apple under Deficit Irrigation

**Riccardo Lo Bianco** 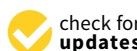

Department of Agricultural, Food and Forest Sciences, University of Palermo, Palermo 90128, Italy;
riccardo.lobianco@unipa.it; Tel.: +39-091-238-96097

**Abstract:** Predicting apple yield in relation to tree water use is important for irrigation planning and evaluation. The aim of the present study was to identify measurable variables related to tree water use that could predict final fruit yield of apple trees under different strategies of deficit irrigation. Adult 'Gala' and 'Fuji' apple trees were exposed to conventional irrigation (CI), delivering 100% of crop evapotranspiration; partial root zone drying (PRD), delivering 50% of CI water only on one alternated side of the root-zone; and continuous deficit irrigation (CDI), delivering 50% of CI water on both sides of the root-zone. Integrals of soil ($SWD_{int}$) and leaf ($LWSD_{int}$) water deficit along with growth and stomatal conductance ($Gs_{int}$) were calculated across each season and used to estimate total conductance ($GS_{tree}$) and transpiration ($Tr_{tree}$) per tree, transpiration efficiency on a fruit ($GR_{fruit}/Tr$) or tree ($GR_{trunk}/Tr$) growth basis, and transpiration productivity ($Yield/Tr_{tree}$). 'Fuji' trees had higher $Yield/Tr_{tree}$, but had lower $GR_{trunk}/Tr$ and similar $GR_{fruit}/Tr$ compared to 'Gala' trees. In 'Fuji', CDI reduced yield, trunk growth, leaf hydration, and gas exchange, while in 'Gala', it did not reduce yield and gas exchange. In 'Fuji', a linear combination of $GR_{trunk}/Tr$, $GR_{fruit}/Tr$, and $Gs_{tree}$ contributed to predicting yield, with $GR_{fruit}/Tr$ explaining nearly 78% of the model variability. In 'Gala', a linear combination of $LWSD_{int}$ and $Gs_{tree}$ contributed to predicting yield, with $Gs_{tree}$ explaining over 79% of the model variability. These results indicate that measuring tree water status or water use may help predict final apple yields only in those cultivars like 'Gala' that cannot limit dehydration by closing stomates because of carbon starvation. In more vigorous cultivars like 'Fuji', transpiration efficiency based on fruit growth can be a powerful predictor of final yields.

**Keywords:** leaf water saturation deficit; partial root zone drying; stomatal conductance; transpiration efficiency; transpiration productivity

## 1. Introduction

On cultivated land, it is estimated that environmental stresses significantly limit agricultural production, and global climate changes are constantly increasing such limitations. Environmental stress tolerance is therefore a critical concern for horticulturists if they hope to increase fruit production as population increases. In particular, plant tissue dehydration (drought stress) may cause direct and indirect decreases in fruit quantity and quality. Indeed, drought may also affect photosynthesis and nutrient uptake causing indirect yield reductions.

A great share of the annual precipitation is lost to evapotranspiration (ET, from 70% up to 90% in arid areas) [1]. This fact proves the importance of adequately estimating the ET component of the hydrologic cycle in predicting on-farm irrigation water management and irrigation planning [2], especially if we consider that without ET there is no production [3]. In particular, the transpiration to ET coefficients have been widely used for precise and efficient irrigation management [4].

Crop yield is determined by both available water quantity and plant water use efficiency [5]. At the physiological level, water use efficiency (WUE) can be defined as the ratio between photosynthesis

and transpiration, also defined as transpiration efficiency or instantaneous WUE. This WUE is quite difficult to monitor at the whole tree scale, and even more at the orchard level. For horticultural evaluations, WUE can be more easily expressed as fruit yield per unit of irrigation water, or irrigation water productivity (IWP). Recently, many studies have focused on IWP as being directly related to the increase of WUE [6]. In this study, the use of transpiration efficiency or productivity based on the ratios between relative trunk or fruit growth and transpiration rate as well as the ratio between fruit yield and total tree transpiration is proposed.

Several factors can cause variations of WUE in plants, e.g., air humidity, the different carboxylation mechanism of C3 and C4 plants and, in the long period, the losses due to respiration and assimilate partitioning. Indeed, it has been widely demonstrated that it is possible to improve plant carbohydrate distribution towards reproductive structures, such as fruits, by keeping the plants in a state of mild water deficit, in this way controlling the excessive vegetative growth [7]. This concept has represented in the last decades the basis for a long list of trials investigating the outcomes of what was called "regulated deficit irrigation" (RDI) by Chalmers et al. [8] or "controlled deficit irrigation" by English [9] and [10]. As a matter of fact, trials conducted on several crops showed that IWP tends to increase with deficit compared to conventional irrigation [11,12].

Increasing IWP has been more successful in trees than in field crops for several reasons [13]. Fruit quality, for example, strongly affects crop value, but is not associated with biomass production and water use. In addition, tree fruit growth may not be sensitive to water deficit in certain periods and developmental stages [14]. This, in combination with low volume/high frequency irrigation systems, gives the best opportunities to manage fruit trees under controlled water deficit.

A specific deficit irrigation practice that has received particular attention in the last decades and seems to achieve significant water savings with limited information inputs from the grower is partial root zone drying (PRD) [15,16]. With PRD, one half of the root system is cyclically left to dry; roots in drying soil produce chemical signals (abscisic acid, cytokinins, pH changes), which are translocated to the shoots [17] where they induce partial stomatal closure, reduce transpiration, and ultimately increase WUE [15]. Thanks to the well-watered half of the root zone, the effect on plant water potential is minimal [18] and other metabolic and physiological processes associated to water stress are not affected [15,19]. This deficit irrigation technique has produced positive outcomes in a number of fruit species, and in apple, numerous PRD studies have reported significant increases of IWP and even yields similar to those of full irrigated trees [16].

For the reasons above, understanding the transpiration mechanisms of plants and the factors affecting final crop yield under water deficit becomes a priority. Green plants have indeed many structures and control devices, which allow them to function efficiently even in rapidly changing environments. At the leaf level, transpiration is controlled by physiological and structural factors with stomatal aperture and conductance assuming a primary role [20]. Stomatal conductance ($g_s$) responds to several factors, such as light, $CO_2$ concentration, vapor pressure deficit, leaf temperature, leaf abscisic acid, and soil water potential. This latter factor influences $g_s$ by a hormonal signal (abscisic acid) originating in the roots, a sort of biological switch when drought occurs [21].

The prediction of apple yield in relation to water requirement or ET is important for irrigation planning and evaluation. Considerable research has led to the development of simple models for predicting mostly yield of field crops from evapotranspiration during the growing season [22–24]. The aim of the present study was to identify measurable variables related to tree water use that could serve for the development of a model to predict final fruit yield of apple trees under deficit irrigation. The same yield predicting variables and models could be useful for fine tuning of deficit irrigation management.

## 2. Materials and Methods

Data of the present study are further calculations and analysis of measurements reported in Lo Bianco and Francaviglia [25]. The study was conducted in 2008 and 2009 near Caltavuturo (37°49′ N

and 850 m a.s.l.), in central Sicily. Plant material consisted of eight-year-old 'Gala' and 'Fuji' apple trees on M.9 rootstock, trained to a central leader, and spaced at 4 m between rows and 1.5 m within rows. Soil type was a sandy clay loam (53.3% sand, 17.6% silt, and 29.1% clay) with pH 7.3 and 1.8% active carbonates, and soil water potential around −17 kPa at field capacity. With the exception of irrigation, all trees received the same cultural practices.

In the field, two nearby rows (one with 36 'Gala' trees, the other with 36 'Fuji' trees) were selected and divided into four blocks, each including three trees per irrigation treatment. Contiguous irrigation treatments on the same row were separated by two buffer trees. In June, three irrigation treatments were imposed: (1) conventional irrigation (CI), delivering 100% of crop evapotranspiration ($ET_c$); (2) PRD, where trees received 50% of CI water only on one alternated side of the root zone; (3) continuous deficit irrigation (CDI), where trees received 50% of CI water on both sides of the root zone. Wet and dry sides of PRD trees were alternated every 2–3 weeks when soil water potential in the dry side reached values of approximately −100 to −150 kPa.

Weather parameters were monitored with a μMetos weather station (Pessl, Austria) positioned within the experimental plot and used to determine reference evapotranspiration ($ET_0$) according to the FAO Penman–Monteith method and crop evapotranspiration ($Et_c$) [26].

Instantaneous vapor pressure deficit (VPD) was calculated from canopy air temperature (in °C) and relative humidity (in %) measured on the same dates and at the same time as stomatal conductance.

Soil water potential was monitored continuously with six Watermark sensors (Irrometer Co., Riverside, CA, USA) directly connected to the weather station. In drip irrigated apple trees, most of the active roots are within the first 60 cm of soil depth. For this reason, Watermark sensors were positioned at a fixed depth of 40 cm and a distance of about 80 cm from emitters and 1 m from the tree trunk in opposite sides of the root-zone. Integrals of soil water deficit ($SWD_{int}$) across each irrigation season and treatment were calculated as:

$$SWD_{int} = \Sigma_{(1..t)} \, | \, (SWD_i - SWD_{FC}) \, | , \tag{1}$$

where t is the number of days in the irrigation season, $SWD_i$ are average daily measures of soil water potential, and $SWD_{FC}$ is soil water potential at field capacity. $SWD_{int}$ was used as an indication of soil water deficit accumulated in the root-zone of each treatment during the irrigation periods.

Every two weeks during the irrigation period, at mid-morning two mature, sun-exposed leaves per tree were collected and transported in ice to the laboratory for determination of fresh weight (FW), turgid weight (TW) after rehydrating leaves for 24 h at 8 °C in the dark, and dry weight (DW) after drying leaves at 60 °C to constant weight. Leaf relative water content (RWC) was calculated as [(FW − DW)/(TW − DW)] × 100. Leaf water saturation deficit (LWSD) was calculated as 1 − (RWC/100) and integrated across the irrigation period using the equation proposed by García-Tejero et al. [27] and modified from Myers [28]:

$$LWSD_{int} = \Sigma_{(1..t)} \, | \, LWD_{i+1} \times (n_{i+1} - n_i) + \frac{1}{2} \, (LWD_i - LWD_{i+1}) \times (n_{i+1} - n_i) \, | , \tag{2}$$

where t is the number of sampling days, $LWD_i$ and $LWD_{i+1}$ are leaf water deficit values measured on two consecutive sampling days (i and i + 1), and $n_{i+1}$ and $n_i$ the days corresponding to two serial samplings. This variable is the integral of a proportion (0 to 1) and therefore can be considered unitless.

On the same dates, stomatal conductance ($g_s$) was measured with an AP4 Delta-T porometer (Delta-T Devices, Cambridge, UK) on two leaves similar to those used for RWC measurements. Stomatal conductance was also integrated across the irrigation period according the following equation:

$$Gs_{int} = \Sigma_{(1..t)} \, | \, g_{i+1} \times (n_{i+1} - n_i) + \frac{1}{2} \, (g_i - g_{i+1}) \times (n_{i+1} - n_i) \, | , \tag{3}$$

where t is the number of sampling days, $g_i$ and $g_{i+1}$ are leaf stomatal conductance values measured on two consecutive sampling days (i and i + 1), and $n_{i+1}$ and $n_i$ the days corresponding to two serial measurements.

In each year, one fruit per tree was measured bi-weekly in size (height and width) with a digital caliper. Relative seasonal fruit growth was calculated as the total increase in average diameter (mm) divided by the initial diameter of the fruit (mm). Trunk circumference was measured at about 15 cm above the graft union at the beginning and end of the two growing seasons. Trunk cross-section area (TCSA) was derived from trunk circumference and taken as an indicator of apple tree size [29]. Tree growth was calculated as the increase in TCSA divided by the initial TCSA. Total leaf area per tree (LA) was destructively measured on a separate set of trees from the two cultivars and related to TCSA by regression analysis. The function obtained was used to estimate LA from TCSA measurements in the trees in trial.

Integrals of leaf transpiration (Tr) were derived from $Gs_{int}$ and VPD as follows:

$$Tr = Gs_{int} \times (VPD/101.3), \tag{4}$$

Where 101.3 is the barometric pressure in kPa at sea level. Integrals of soil ($SWD_{int}$) and leaf ($LWSD_{int}$) water deficit along with growth and stomatal conductance ($Gs_{int}$) were calculated across each season and used to estimate total conductance ($GS_{tree}$) and transpiration ($Tr_{tree}$) per tree, transpiration efficiency on a fruit ($GR_{fruit}/Tr$) or tree ($GR_{trunk}/Tr$) growth basis, and transpiration productivity ($Yield/Tr_{tree}$). Transpiration efficiency on a per tree growth basis was estimated by dividing trunk growth by Tr ($Gr_{trunk}/Tr$), transpiration efficiency on a fruit growth basis was estimated by dividing fruit growth by Tr ($Gr_{fruit}/Tr$), while transpiration productivity was obtained by dividing yield by $Tr_{tree}$ ($Yield/Tr_{tree}$). Transpiration productivity is a very useful measure which is more accurate than IWP and more practical than instantaneous WUE as a trait for improving fruit productivity under limited water resources. Total stomatal conductance per tree was estimated from $Gs_{int}$ and LA, while total transpiration per tree was estimated from Tr and LA.

Data were tested for normal distribution and equal variances and analyzed by analysis of variance and regression procedures using Systat and SigmaPlot software (Systat Software Inc., Richmond, CA, USA). Least squares multiple linear regression with a backward stepwise technique was used to find the best set of variables predicting final apple yield. Means were separated by Tukey's multiple comparison test at *P* < 0.05.

## 3. Results and Discussion

The relationship between TCSA and LA was described by a non-linear polynomial function, as shown in Figure 1. Canopy and root system size have been linearly related to TCSA in apple [29,30]. In this study with eight-year-old apple trees, the non-linear relationship between TCSA and LA can be explained by canopy size constraints imposed by planting density, tree training form, and pruning. In other words, more vigorous trees (e.g., 'Fuji' trees) were pruned more heavily than weaker trees to remain within the allotted space and avoid competition for light. In this way, canopy and leaf area of trees with different TCSA are brought back to similar sizes determining the observed non-linear relationship between TCSA and LA.

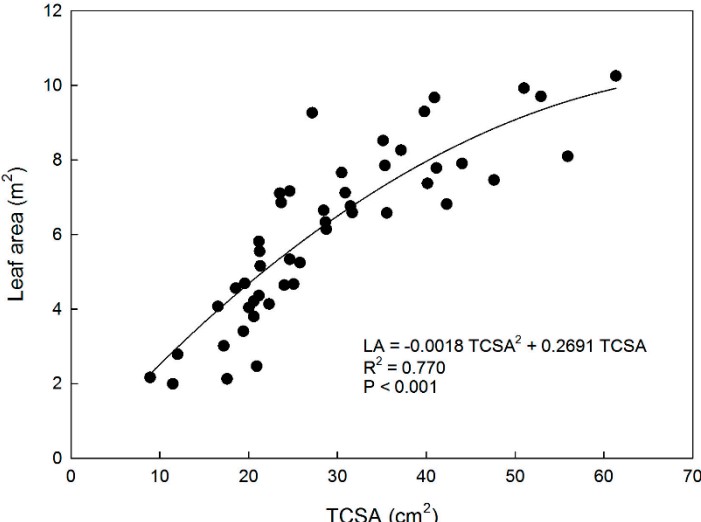

**Figure 1.** Relationship between trunk cross-section area (TCSA) and total leaf area (LA) in eight-year-old 'Gala' and 'Fuji' apple trees grafted on M.9 rootstock, trained to a central leader, spaced at 4 × 1.5 m, and grown near Caltavuturo, Sicily.

The imposed irrigation treatments effectively determined the expected differences in soil water deficit. Indeed, on average of the two seasons, $SWD_{int}$ was four times higher in CDI (5.62 MPa) than in CI (1.43 MPa) trees; despite the same irrigation volumes, PRD reported intermediate $SWD_{int}$ (4.61 MPa) with 22% lower values than CDI trees. This has been attributed to greater wetted soil surface and consequent soil evaporation in CDI than in PRD in previous studies [25,31,32].

As expected and regardless of irrigation strategy, 'Fuji' trees were bigger, had greater LA, transpired more water, and yielded more fruit than 'Gala' trees, as shown in Tables 1 and 2. On the other hand, transpiration efficiency in terms of whole tree growth was higher in 'Gala' than in 'Fuji' trees; specifically, 'Gala' trees had higher $GR_{trunk}/Tr$ but similar $GR_{fruit}/Tr$ and lower $Yield/Tr_{tree}$ compared to 'Fuji' trees, as shown in Tables 1 and 2, suggesting that they partitioned assimilates mostly to vegetative rather than fruit growth. This may be at least in part due to their smaller size and fewer constraints to acquire soil resources and fill the allotted space compared to 'Fuji' trees.

The two apple cultivars responded differently to soil water deficit and irrigation strategy. In 'Fuji', PRD maintained plant gas exchange, hydration levels, growth, and productivity similar to CI, while CDI induced significant reductions of yield, tree growth, leaf hydration, and gas exchange, as shown in Table 1. In contrast, no yield and gas exchange differences were observed in deficit irrigated 'Gala' trees, while CDI decreased leaf hydration levels and tree growth compared to both CI and PRD, as shown in Table 2. Transpiration productivity was significantly increased by PRD mainly in 'Fuji' while it was reduced by CDI in 'Gala'. Given the milder soil and leaf water deficit induced by PRD compared to CDI, the different responses of the two cultivars to irrigation may be associated with different levels of water stress resistance, with 'Gala' exhibiting lower ability to limit dehydration than 'Fuji'. In other words, both cultivars tend to close stomates and transpire less as soil water deficit progresses, minimizing symptoms of leaf dehydration (isohydric behavior). Yet, 'Gala' showed higher $LWSD_{int}$ (significant leaf dehydration under both PRD and CDI) than 'Fuji' (leaf dehydration only under CDI). This is at least in part due to the larger 'Fuji' root systems which were able to explore more and deeper soil layers and acquire more water and nutrients than 'Gala' roots.

**Table 1.** Yield, growth (GR), and water-use related variables in 'Fuji' apple trees under conventional irrigation (CI), partial root zone drying (PRD), and continuous deficit irrigation (CDI).

| | CI | | PRD | | CDI | | |
|---|---|---|---|---|---|---|---|
| Yield (kg/tree) | 36.3 | ab [z] | 39.4 | a | 33.5 | b | * [y] |
| Yield Efficiency (kg cm$^{-2}$) | 0.729 | ab | 0.853 | a | 0.607 | b | ** |
| GR$_{trunk}$ (cm$^2$ cm$^{-2}$) | 0.051 | a | 0.052 | a | 0.037 | b | * |
| GR$_{fruit}$ (mm mm$^{-1}$) | 0.541 | | 0.545 | | 0.501 | | ns |
| Leaf Area (m$^2$) | 9.02 | | 8.70 | | 9.39 | | ns |
| Vapor Pressure Deficit (kPa) | 158 | | 159 | | 159 | | ns |
| LWSD$_{int}$ [x] | 10.1 | a | 10.5 | ab | 11.1 | b | * |
| Gs$_{int}$ [w] (mol m$^{-2}$ s$^{-1}$) | 16.0 | a | 15.5 | a | 12.5 | b | ** |
| Tr [v] (mol m$^{-2}$ s$^{-1}$) | 25.6 | a | 24.9 | a | 20.0 | b | ** |
| GR$_{trunk}$/Tr | 0.022 | | 0.023 | | 0.019 | | ns |
| GR$_{fruit}$/Tr | 0.025 | | 0.025 | | 0.028 | | ns |
| Yield/Tr$_{tree}$ | 0.177 | b | 0.210 | a | 0.191 | ab | * |
| Tr$_{tree}$ [u] (mol s$^{-1}$/tree) | 229 | | 216 | | 188 | | ns |
| Gs$_{tree}$ [u] (mol s$^{-1}$/tree) | 143 | | 136 | | 118 | | ns |

[z] Mean separation within rows by Tukey's multiple comparison test at *P* < 0.05
[y] Level of statistical significance for irrigation factor from analysis of variance: ns, *P* > 0.05; *, *P* < 0.05; **, *P* < 0.01; ***, *P* < 0.001.
[x] LWSD$_{int}$ is leaf water saturation deficit integrated across the irrigation period.
[w] Gs$_{int}$ is stomatal conductance integrated across the irrigation period.
[v] Tr is transpiration integrated across the irrigation period.
[u] Seasonal integrals of total conductance (GS$_{tree}$) and transpiration (Tr$_{tree}$) per tree.

**Table 2.** Yield, growth (GR), and water-use related variables in 'Gala' apple trees under conventional irrigation (CI), partial root zone drying (PRD), and continuous deficit irrigation (CDI).

| | CI | | PRD | | CDI | | |
|---|---|---|---|---|---|---|---|
| Yield (kg/tree) | 19.3 | | 18.5 | | 17.2 | | ns [z] |
| $Yield_{Eff}$ (kg cm$^{-2}$) | 0.600 | | 0.659 | | 0.537 | | ns |
| $GR_{trunk}$ (cm$^2$ cm$^{-2}$) | 0.082 | a [y] | 0.071 | a | 0.053 | b | *** |
| $GR_{fruit}$ (mm mm$^{-1}$) | 0.545 | | 0.568 | | 0.563 | | ns |
| Leaf Area (m$^2$) | 6.69 | | 6.31 | | 6.95 | | ns |
| Vapor Pressure Deficit (kPa) | 169 | | 168 | | 170 | | ns |
| $LWSD_{int}$ [x] | 7.53 | a | 8.28 | ab | 8.93 | b | * |
| $Gs_{int}$ [w] (mol m$^{-2}$ s$^{-1}$) | 13.9 | | 12.9 | | 12.4 | | ns |
| $Tr$ [v] (mol m$^{-2}$ s$^{-1}$) | 23.7 | | 21.0 | | 22.0 | | ns |
| $GR_{trunk}/Tr$ | 0.044 | a | 0.044 | a | 0.029 | b | * |
| $Gr_{fruit}/Tr$ | 0.028 | | 0.033 | | 0.028 | | ns |
| $Yield/Tr_{tree}$ | 0.141 | ab | 0.161 | a | 0.124 | b | * |
| $Tr_{tree}$ [u] (mol s$^{-1}$/tree) | 154 | | 131 | | 150 | | ns |
| $Gs_{tree}$ [u] (mol s$^{-1}$/tree) | 90.5 | | 77.4 | | 88.5 | | ns |

[z]　　Level of statistical significance for irrigation factor from analysis of variance: ns, $P > 0.05$; *, $P < 0.05$; **, $P < 0.01$; ***, $P < 0.001$.

[y]　　Mean separation within rows by Tukey's multiple comparison test at $P < 0.05$

[x]　　$LWSD_{int}$ is leaf water saturation deficit integrated across the irrigation period.

[w]　　$Gs_{int}$ is stomatal conductance integrated across the irrigation period.

[v]　　$Tr$ is transpiration integrated across the irrigation period.

[u]　　Seasonal integrals of total conductance ($GS_{tree}$) and transpiration ($Tr_{tree}$) per tree.

In 'Fuji', a linear combination of $GR_{trunk}/Tr$, $GR_{fruit}/Tr$, and $Gs_{tree}$ contributed to predicting yield of apple trees under soil water deficit, as shown in Table 3. In this cultivar, $GR_{fruit}/Tr$ was the most important variable for predicting yield, explaining nearly 78% of the model variability, while $GR_{trunk}/Tr$ can be considered negligible as it explained only about 3% of the model variability, as shown in Table 3. On the other hand, a linear combination of $LWSD_{int}$ and $Gs_{tree}$ contributed to predicting yield of 'Gala' apple trees under soil water deficit, as shown in Table 4. In this cultivar, $Gs_{tree}$ was the most important variable for predicting yield, explaining over 79% of the model variability, as shown in Table 4. This difference between the two cultivars indicates that fruit yield of 'Gala' trees was more sensitive to stomatal closure compared to 'Fuji' trees. This may be due to differences in tree size and leaf water deficit levels. The increase of $LWSD_{int}$ over the control was indeed greater in 'Gala' (0.75 and 1.4 for PRD and CDI, respectively) than in 'Fuji' (0.4 and 1 for PRD and CDI, respectively). The effect of tree water status on apple yield has been already documented, although other factors like crop load may have stronger effects than water deficit on yield [33]. In addition, the larger 'Fuji' trees may have been less sensitive to stomatal closure than 'Gala' trees because of greater carbon and water storage in permanent structures. In this regard, others have reported contrasting results indicating positive or no effect of tree size and capacitance on water status [34,35], while there is little doubt about the role of permanent structures as carbon reservoirs. In addition to carbon reserves in permanent structures, differences in LA and photosynthetic rates as well as nutrient acquisition may play a significant role in the response of the two cultivars. In the present study, differences in tree size and carbon and water storage may also help explain the higher transpiration productivity in 'Fuji' than in 'Gala' and the major contribution of $Gr_{fruit}/Tr$ to yield prediction in 'Fuji'. $LWSD_{int}$ was a relatively weak yield predictor only in 'Gala', suggesting that even under soil water limiting conditions, factors other than water (e.g., assimilation rate, nutrient status, flower fertility, pollination)

are major determinants of apple fruit yield formation and a simple measurement of tree water status may not serve as a solid yield predictor.

**Table 3.** Multiple linear regression model and parameters contributing to predict yield in 'Fuji' apple trees under deficit irrigation.

| Yield = 28.5 + (2774 × GR_trunk/Tr) − (295 × GR_fruit/Tr) + (0.070 × Gs_tree) | | | | | |
|---|---|---|---|---|---|
| **N = 65** | **R = 0.554** | **R$^2$ = 0.306** | **SE of Estimate = 8.85** | ***P* < 0.001** | |
| **Parameters** | **Coefficient** | **SE** | **t** | ***P*** | **% of SSreg** |
| Constant | 28.5 | 6.74 | 4.23 | <0.001 | - |
| GR$_{trunk}$/Tr $^z$ | 2774 | 973 | 2.85 | 0.006 | 3.1 |
| GR$_{fruit}$/Tr $^y$ | −295 | 118 | −2.51 | 0.015 | 77.9 |
| Gs$_{tree}$ $^x$ | 0.070 | 0.031 | 2.26 | 0.027 | 19.0 |

$^z$　　Trunk growth/transpiration.
$^y$　　Fruit growth/transpiration.
$^x$　　Seasonal integral of total conductance (Gs) per tree.

**Table 4.** Multiple linear regression model and parameters contributing to predict yield in 'Gala' apple trees under deficit irrigation.

| Yield = −0.019 + (1.35 × LWSD_int) + (0.085 × Gs_tree) | | | | | |
|---|---|---|---|---|---|
| **N = 68** | **R = 0.410** | **R$^2$ = 0.168** | **SE of Estimate = 6.33** | ***P* = 0.003** | |
| **Parameters** | **Coefficient** | **SE** | **t** | ***P*** | **% of SSreg** |
| Constant | −0.019 | 5.50 | −0.003 | 0.997 | - |
| LWSD$_{int}$ $^z$ | 1.35 | 0.52 | 2.59 | 0.012 | 20.5 |
| Gs$_{tree}$ $^y$ | 0.085 | 0.026 | 3.23 | 0.002 | 79.5 |

$^z$　　LWSD$_{int}$ is leaf water saturation deficit integrated across the irrigation period.
$^y$　　Seasonal integral of total conductance (Gs) per tree.

In conclusion, the more vigorous 'Fuji' trees were more efficient than 'Gala' trees under soil water deficits in terms of yield and transpiration productivity. Our results indicate that measuring tree water status or gas exchange may help predict final apple yields only in those trees and cultivars (like 'Gala' in this study) that are not able to limit dehydration by closing stomates because of carbon starvation. In more vigorous trees and cultivars like 'Fuji', transpiration (or water use) efficiency towards fruit growth seems to be a powerful predictor of final yields.

**Funding:** This research received no external funding.

**Conflicts of Interest:** The author declares no conflict of interest.

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
