# Peer review of "Water-Related Variables for Predicting Yield of Apple under Deficit Irrigation"

_horticulturae, doi:10.3390/horticulturae5010008_

Round 1

Reviewer 1 Report

Introduction should be improved : ETc*kc approach valid only for full surface irrigation ; intrinsic WUE is An/gs and not An/E. ABA biosynthesis is not restricted inly to the root system

Material and methods : measurement of soil water potential only at 40cm is a minimum, and should be discussed in regard to the root system distribution. Integration of leaf relative water content and stomatal conductance is an interesting idea.

Figure 1 is a source of questioning : haow can such a variability between 8-y old trees be observed?

Table 1 & 2 : interesting panel of integrative variables

Tables 3 & 4 : a courageous trial or predicting yield through a multiple linear regression, but

1/ units are lacking.

2/ the reader has quite no idea of method of optimization of the relation

3/ yield prediction in Gala is rather tricky

4/ what practical use for the growers ?

Author Response

Thanks for the constructive suggestions. A point by point answer to your comments follows:

Introduction should be improved :

ETc*kc approach valid only for full surface irrigation ;

Sorry, I don't see anything relating to ETc*kc in the introduction

intrinsic WUE is An/gs and not An/E.

corrected, "intrinsic" was supposed to be "instantaneous"

ABA biosynthesis is not restricted inly to the root system

added a brief mention to leaf ABA

Material and methods : measurement of soil water potential only at 40cm is a minimum, and should be discussed in regard to the root system distribution.

sentence added

Figure 1 is a source of questioning : haow can such a variability between 8-y old trees be observed?

the total TCSA variability visible from the x-axis of fig. 1 is about half (10-35) associated to Gala trees and the other half (35-60) to Fuji trees. In terms of trunk circumference, ranges are about 12-20 cm for Gala and 20-28 cm for Fuji; after 8 years of growing at slightly different rates, I guess these can be considered acceptable differences; besides, those LA/TCSA measurements were conducted on trees outside the trial plot and from different spots in the orchard; indeed the range of variability of fig. 1 is greater than the one observed in the trees in trial

Table 1 & 2 : interesting panel of integrative variables

Tables 3 & 4 : a courageous trial or predicting yield through a multiple linear regression, but 

1/ units are lacking.

same units as in Tables 1 and 2

2/ the reader has quite no idea of method of optimization of the relation

added a sentence at the end of M&M; I hope that is what you meant

3/ yield prediction in Gala is rather tricky

I am not sure I understand what you mean here

4/ what practical use for the growers ?

if the conclusions of the study are valid, then growers can find some useful information on what to look for or after when irrigation water is an issue

Reviewer 2 Report

Dear Editor and author, I have finished the review of the ms and I find it well written and the calculations made have some interest. Nonetheless, the findings are not completely solving the aims pursued.

I have a few suggestions I would like author to consider:

Line 9 requirement? As author knows well, plant water requirements are determined and can be satisfied or not. In this experimentation, requirements were often not satisfied since plants are subjected to different strategies of DI. So, I suggest change water requirement by water use.

L12 under different strategies of deficit irrigation

L43 Of course crop yield is determined by many more things than irrigation, flowering level, pollination success, nutrition, pest damages and etc.. This is not saying irrigation is not important is putting results in perspective. Perhaps some of these factors limited yield (nothing was said at this regard), and maybe in a different extent to the different genotypes and for that reason plat response to water deficit seemed different. Please address these points if you can in the Discussion as well. See Line 235.

L153 When was Trunk cross sectional area measured? At the end of the season? Then trunk growth increase is more a result more than a cause. If before bloom, it could be a predictor. Please explain.

L167-174. Justify paragraph

L187-190. As before, some parameters can be predictors or results of. For instance if you have less fruit load you have more exuberant vegetative growth in that season, but not the contrary. Although that larger canopy may facilitate in turn subsequently larger yield IF PRUNING DOES NOT LIMIT CANOPY SIZE what use to be the practice in adult trees. So, be careful when explaining relationships not always as in this case a causal relation.  

Line 194. Table 1

Line 196. Table 2.

Table 1 and 2. I would like to see mean separation. The existence of significant differences but not showing the separation of means makes readers do mental calculations about mean ± Standard error?

L 221 it could be interesting knowing if lower yield was due to smaller or less fruits. Since fruit thinning supposedly was performed, as I assume crop load was regulated, then, the effect should be more expressed on fruit size. Please explain, because this information is useful.   

Table 3. Omit GRtrunk/Tr since in the regression equation explains very little of the variability observed and makes equation more complex.

Author Response

First of all, let me thank you for the constructive suggestions. Here is a point by point answer to your comments:

Line 9 requirement? As author knows well, plant water requirements are determined and can be satisfied or not. In this experimentation, requirements were often not satisfied since plants are subjected to different strategies of DI. So, I suggest change water requirement by water use.

ok, done

L12 under different strategies of deficit irrigation

ok, done

L43 Of course crop yield is determined by many more things than irrigation, flowering level, pollination success, nutrition, pest damages and etc.. This is not saying irrigation is not important is putting results in perspective. Perhaps some of these factors limited yield (nothing was said at this regard), and maybe in a different extent to the different genotypes and for that reason plat response to water deficit seemed different. Please address these points if you can in the Discussion as well. See Line 235.

added

L153 When was Trunk cross sectional area measured? At the end of the season? Then trunk growth increase is more a result more than a cause. If before bloom, it could be a predictor. Please explain.

TCSA was measured before budbreak and at leaf fall in each season (this was already mentioned in the paper); what I used as a predictor is not TCSA per se but trunk growth [(TCSAend - TCSAinit)/TCSAinit]; I guess trunk growth could be measured at any time during fruit growth and be used as a predictor, it depends on when we want to do predictions (of course early predictions are never the most accurate)

L167-174. Justify paragraph

ok, done

L187-190. As before, some parameters can be predictors or results of. For instance if you have less fruit load you have more exuberant vegetative growth in that season, but not the contrary. Although that larger canopy may facilitate in turn subsequently larger yield IF PRUNING DOES NOT LIMIT CANOPY SIZE what use to be the practice in adult trees. So, be careful when explaining relationships not always as in this case a causal relation.  

true, but if you consider this is the integration of measurements of two consecutive seasons... I think that my explanations should still hold, for the most part at least

Line 194. Table 1

ok, done

Line 196. Table 2.

ok, done

Table 1 and 2. I would like to see mean separation. The existence of significant differences but not showing the separation of means makes readers do mental calculations about mean ± Standard error?

ok, done (some journals don't like it, but hopefully it will be fine here)

L 221 it could be interesting knowing if lower yield was due to smaller or less fruits. Since fruit thinning supposedly was performed, as I assume crop load was regulated, then, the effect should be more expressed on fruit size. Please explain, because this information is useful.  

No difference in fruit size, only in crop load due to fruit drop. You can find this info in:

D. FRANCAVIGLIA, V. FARINA, G. AVELLONE and R. LO BIANCO. 2013. Fruit yield and quality responses of apple cvars Gala and Fuji to partial rootzone drying under Mediterranean conditions. The Journal of Agricultural Science, 151, 556–569 doi:10.1017/S0021859612000718

Table 3. Omit GRtrunk/Tr since in the regression equation explains very little of the variability observed and makes equation more complex.

the choice of retaining or deleting a variable is linked to statistics, GRtrunk/Tr is significant and I think we need to keep it in the model; I added a sentence pointing out it explains a negligible fraction of the model variability; I also added a better explanation of the regression procedure to recognize useful variables for the model (right at the end of M&M)